# S100A4 Is a Biomarker of Tumorigenesis, EMT, Invasion, and Colonization of Host Organs in Experimental Malignant Mesothelioma

**DOI:** 10.3390/cancers12040939

**Published:** 2020-04-10

**Authors:** Joëlle S. Nader, Jordan Guillon, Coralie Petit, Alice Boissard, Florence Franconi, Stéphanie Blandin, Sylvia Lambot, Marc Grégoire, Véronique Verrièle, Béatrice Nawrocki-Raby, Philippe Birembaut, Olivier Coqueret, Catherine Guette, Daniel L. Pouliquen

**Affiliations:** 1Université de Nantes, Inserm, CRCINA, F-44000 Nantes, France; joelle03nader@gmail.com (J.S.N.); sylvia.lambot@inserm.fr (S.L.); 2Université d’Angers, Inserm, CRCINA, F-44000 Nantes, France; jordanguillon5@gmail.com (J.G.); coralie.petit49@gmail.com (C.P.); olivier.coqueret@univ-angers.fr (O.C.); 3Université d’Angers, ICO Cancer Center, Inserm, CRCINA, F-44000 Nantes, France; alice.boissard@ico.unicancer.fr (A.B.); veronique.verriele@ico.unicancer.fr (V.V.); catherine.guette@ico.unicancer.fr (C.G.); 4Université d’Angers, PRISM Plate-forme de Recherche en imagerie et spectroscopie multi-modales, F-49000 Angers, France; florence.franconi@univ-angers.fr; 5Université de Nantes, Plate-forme MicroPICell, SFR François Bonamy, F-44000 Nantes, France; stephanie.blandin@univ-nantes.fr; 6Université de Nantes, Inserm, CRCINA, SIRIC ILIAD, F-44000 Nantes, France; marc.gregoire@inserm.fr; 7Université de Reims Champagne Ardenne, Inserm, P3Cell UMR-S 1250, SFR CAP SANTE, F-51092 Reims, France; beatrice.raby@univ-reims.fr; 8CHU de Reims, Hôpital Maison Blanche, Laboratoire de Pathologie, F-51092 Reims, France; pbirembaut@chu-reims.fr

**Keywords:** S100A4, malignant mesothelioma, EMT, invasiveness, colonization, proteomics, SWATH-MS, biomarkers, rat tumor model

## Abstract

Recent findings suggest that S100A4, a protein involved in communication between stromal cells and cancer cells, could be more involved than previously expected in cancer invasiveness. To investigate its cumulative value in the multistep process of the pathogenesis of malignant mesothelioma (MM), SWATH-MS (sequential window acquisition of all theoretical fragmentation spectra), an advanced and robust technique of quantitative proteomics, was used to analyze a collection of 26 preneoplastic and neoplastic rat mesothelial cell lines and models of MM with increasing invasiveness. Secondly, proteomic and histological analyses were conducted on formalin-fixed paraffin-embedded sections of liver metastases vs. primary tumor, and spleen from tumor-bearing rats vs. controls in the most invasive MM model. We found that S100A4, along with 12 other biomarkers, differentiated neoplastic from preneoplastic mesothelial cell lines, and invasive vs. non-invasive tumor cells in vitro, and MM tumors in vivo. Additionally, S100A4 was the only protein differentiating preneoplastic mesothelial cell lines with sarcomatoid vs. epithelioid morphology in relation to EMT (epithelial-to-mesenchymal transition). Finally, S100A4 was the most significantly increased biomarker in liver metastases vs. primary tumor, and in the spleen colonized by MM cells. Overall, we showed that S100A4 was the only protein that showed increased abundance in all situations, highlighting its crucial role in all stages of MM pathogenesis.

## 1. Introduction

Identifying key molecular components and the sequence of events that occurs is a crucial step in understanding the tumor metastasis process. For this purpose, a number of proteins were selected, of which S100A4, an important member of the S100 protein family, actively participates in tumor progression and metastasis in various malignant tumors [1]. Moreover, in the context of a pre-metastatic niche [2], S100A4 production serves as a link between inflammation and tumor metastasis, and is indicative of poor prognosis [3].

The acquisition of a metastatic phenotype is associated with the re-activation of the epithelial-to-mesenchymal transition (EMT) [4], which also involves the hypoxia-inducible factor HIF-1 [5]. The contribution of S100A4 to the EMT process has been demonstrated in patients, being particularly highly expressed in the peripheral leading edge of breast cancer [6], and experimentally in a context of transformation of a non-metastatic human prostate cancer cell line, which leads to the cancers’ acquiring invasive properties [7]. The combined gain of S100A4 and loss of membrane E-cadherin in cervical cancer tends to confirm its link with an unfavorable prognosis [8]. Moreover, the role of this biomarker as a central node in a molecular network controlling stemness and EMT has been reported in one of the most aggressive tumors, glioblastoma [9], providing the initial parts of an answer to the question of the relationship between these two processes [10].

Malignant mesothelioma (MM) has one of the worst clinical outcomes, especially its sarcomatoid histological subtype, because of the complex biology of this cancer due to its mesenchymal “pluripotent” origin. Moreover, EMT appears to represent a permanent feature of the sarcomatoid subtype, with a deleterious impact on its prognosis [11]. These peculiarities explain the very modest success of the various therapies applied [12,13], and the challenge of diagnosis and prognosis of this pathology. For these reasons, further studies are required in order to fully understand MM molecular pathogenesis and to develop new target agents [14], particularly for the sarcomatoid subtype, for which current biomarkers are not valid [15]. To fulfill the need for new biomarkers that are effective for MM [16], our team previously demonstrated that SWATH-MS (sequential window acquisition of all theoretical fragmentation spectra), an advanced and robust quantitative proteomics technique, can be applied to the analysis of experimental MM tumor specimens to characterize increasing stages of invasiveness [17].

In this study, our aim was to determine whether S100A4, one of the most important biomarkers for invasiveness, previously identified in a list of 137 proteins [17], was involved in all stages of MM pathogenesis, including tumorigenesis, EMT, invasion, and colonization of host organs, and whether it presented a comparable evolution during the whole process when used alone or when combined with other proteins of interest. Our experimental approach was based on the use of a large biocollection of rat cell lines and MM tumor models [18]. Rat models of MM have been successfully used for decades to decipher the different stages of MM molecular pathogenesis [19]. The immunocompetent inbred F344 rat strain F344 is a good experimental base for designing syngeneic orthotopic tumor models, which are preferable when considering tumor versus immune cell interactions [17,20], and when evaluating the therapeutic efficacy of new drugs [21]. Moreover, this type of experimental model allows MRI monitoring [22]. We used such a model so as to benefit from its genetically identical base, as all individuals bear the same Major Histocompatibility Complex (MHC) haplotype in the RT1 region (RT1^lv^), a situation which potentially limits the source of potential variation in the results when contrasting with humans, while mimicking some of the worst clinical conditions encountered in patients.

## 2. Results

### 2.1. S100A4, a Biomarker for Neoplastic Transformation, EMT, and In Vitro Invasiveness

During the first step, SWATH-MS analysis of the whole biocollection of rat mesothelial cell lines detected 1661 proteins in each cell line. The number of proteins with significantly different abundances, allowing to differentiate between the groups of neoplastic cell lines (C3) and preneoplastic cell lines with epithelioid (C1) or sarcomatoid morphology (C2) [18], was 863 and 856 proteins, respectively. Crossing the two files produced 595 common proteins representing potential biomarkers for neoplastic transformation (Figure 1A). Secondly, this tumorigenesis list of 595 proteins was crossed with an invasiveness list consisting of 251 proteins, presenting significantly different abundances between the three invasive neoplastic cell lines (F4-T2, F5-T1, and M5-T1) (C4) relative to the non-invasive M5-T2 cell line (C5) (Figure 1A). This produced 90 proteins representing potential biomarkers for both neoplastic transformation and in vitro invasiveness (Figure 1B). In parallel, two lists of biomarkers differentiating the two groups of preneoplastic cell lines with sarcomatoid, PNsarc (C2), and epithelioid, PN-[Epith] (C1) morphologies [18], and the two subgroups, PNsarc2 and PNsarc1 (Table 1), were established, consisting of 674 and 192 proteins, respectively. Crossing these two lists with the 90 proteins described above produced one single protein, S100A4 (Figure 1B), that showed an increased abundance between the two groups of preneoplastic cell lines (Figure 1C). This result appeared to be related to EMT, as a significant rise in the abundance of fibronectin and vimentin, two important markers of this process [23], was concomitantly observed (Appendix A).

Given the suggested role of oxidative stress in mesothelial carcinogenesis by promoting EMT processes, particularly with the involvement of *Hif1a* [24], an investigation of its expression profile and 14 other genes coding for various factors (*Ccl3*, *Ccl5*, *Ccl7*, *Egf*, *Erbb2*, *Fgf2*, *Fhit*, *Il1b*, *Il6*, *Nfkb1*, *Pdgfa*, *Stat3*, *Tnf*, and *Vegfa*) [25] revealed that the increased expression of *Hif1a* in PNsarc2 versus PNsarc1 (Table 1) was linked to the expression of *Vegfa*, *CCl7*, and *Stat3* (Figure 2).

### 2.2. S100A4, Biomarker of In Vivo Invasiveness

As a third step, the previous list of 90 proteins that characterized both neoplastic transformation and in vitro invasiveness in cells was crossed with a new list of 206 proteins (Figure 1A) that differentiated the three invasive MM tumors from the non-invasive one. This produced a reduced list of 39 potential biomarkers for both neoplastic transformation and in vitro and in vivo invasiveness (Figure 1B). Finally, the four analyses produced 13 proteins (Figure 1B), for which quantitative changes corresponded to increases or decreases in all situations (Table 2). For these 13 proteins, additional information was collected, such as the mean differences observed not only between groups, but also between subgroups of preneoplastic cells, or between individual invasive neoplastic cell lines or tumors versus the non-invasive ones. Among these 13 proteins, S100A4 was the only one to exhibit a continuous and significant increase in abundance from preneoplastic cell lines with epithelioid and sarcomatoid morphology to neoplastic cells, and from non-invasive to invasive MM cells and tumors (Figure 1C). Five proteins presented a common increase in abundance: Annexin A5, prelamin-A/C, barrier-to-autointegration factor, histone H2A.J, and glutathione reductase (Appendix A). Seven proteins showed a common decrease in abundance: Septin-2, dihydropyrimidinase-related protein 2, myosin-10, the mitochondrial aconitate hydratase, leucine zipper protein 1, echinoderm microtubule-associated protein-like 2, and 5-oxoprolinase (Appendix A).

To evaluate a possible human MM application deriving from these findings, we conducted a preliminary investigation on two tumor samples from the Tumor Bank of the Reims University Hospital Biological Resources, Collection n° AC-2019-340, declared at the Ministry of Health, according to the French Law for the use of tissue samples for research. A comparative SWATH-MS analysis of two paraffin-embedded human MM tumor pieces (sarcomatoid vs. epithelioid, Appendix A) revealed a significant increase in S100A4 abundance in both the tumor and its periphery, and concomitantly in three EMT markers (Appendix A). These results also agreed with our observations on rat preneoplastic cell lines (see Figure 1C, Section 2.1, and Appendix A) and rat MM tumors (Figure 1C, Appendix A, bottom graphs), except for vimentin, due to the fact that all four rat MM tumors presented a sarcomatoid morphology [17]. This role of S100A4 in human MM invasiveness is also consistent with the observation of a decreased survival probability for patients with renal cancer and high S100A4 expression (Appendix A [26]).

### 2.3. S100A4 Increases with the Development Stage of MM Liver Metastases

We previously identified important differences in invasiveness level and stroma composition between our three experimental models of rat invasive MM [17]. In particular, the most invasive, M5-T1, was characterized by the capacity to generate metastases in multiple secondary organs, especially the liver [27], as confirmed by MRI (magnetic resonance imaging) (Figure 3).

We therefore wondered whether the level of S100A4 would change in liver metastases according to their development stage. For this purpose, M5-T1 liver metastases were collected from two groups of rats that differed in the number/extent of secondary organs affected by M5-T1 cell invasion. The first group (advanced stage) corresponded to rats with the initial stages of diaphragm invasion, preservation of the splenic capsule, no nodules infiltrating the peritoneum, and liver metastases of an average of 2.87 mm in diameter, without deep infiltration. The second group (final stage) was characterized by deep invasion of the diaphragm, portal space, and spleen, the presence of nodules infiltrating the peritoneum, and multiple, deep liver metastases of an average of 5.25 mm in diameter. Histological examination revealed two observations that characterized the final stage: the presence of multiple isolated M5-T1 cells deeply infiltrating the organ at distance from the tumor front, and evidence of alterations of the liver parenchyma architecture (Figure 4A, right column). In contrast, the advanced stage was characterized by the absence of these two elements (Figure 4A, left column). SWATH-MS analysis of the two groups of tumors showed that they differed by 148 proteins presenting significant differences in abundance (Figure 1A), 10 of which were also present on the list of 39 biomarkers established in Table 2. Interestingly, the most significant ratios were observed for S100A4 and two enzymes of the glycolysis pathway, triose phosphate isomerase (TPIS) and phosphoglycerate mutase 1 (PGAM1) (Figure 4B).

### 2.4. S100A4, Biomarker for Colonization of the Spleen by MM Tumor Cells

We next determined which of the 39 biomarkers (Table 2) were detectable by SWATH-MS analysis and to what extent their abundances were impacted when colonization by the most invasive M5-T1 cells extended to a central lymphoid organ, the spleen (Figure 1A), in comparison with control rats, and how these changes evolved within the different tumor progression stages. This experiment was justified by the observation of changes in MR images (Figure 3), and histology of the spleen (Figure 5) in tumor-bearing rats relative to normal rats.

Histological examination at high magnification (×800) of a set of paraformaldehyde-fixed paraffin-embedded organs and tissues (liver, pancreas, spleen, kidney, diaphragm, peritoneal wall, lymph nodes, colon, duodenum, omentum, and adipose tissue) collected from 20 M5-T1 tumor-bearing rats at different stages of progression allowed the localization and morphological features of colonizing tumor cells to be determined. The typical features of M5-T1 metastatic cells colonizing different organs are shown in Appendix A.

Next, histological examination of spleen sections from the group of M5-T1 tumor-bearing rats allowed us to identify three different stages of tumor progression according to the presence of tumor cells attached to/crossing the capsule (Figure 5A), and composition/morphological features of cells present in the red pulp (Figure 5B) or periarteriolar lymphoid sheath (Figure 5C). Sections of paraffin-embedded spleens of 20 μm thick from 10 M5-T1 tumor-bearing rats and 10 control rats were prepared for SWATH-MS analysis and biomarker abundances were compared. Table 3 summarizes the results, highlighting the significant changes observed for eight candidate biomarkers of pre-metastatic niche formation, and the fact that S100A4 showed the most significant increase in abundance as a consequence of tumor progression (Figure 6A). Comparison of the abundances for other proteins of the S100 family demonstrated that five proteins were detected in the spleen tissue (S100A11, S100A6, S100A8, S100A9, and S100A4, in increasing order of abundance), S100A9 being the only one that increased (Figure 6B). In contrast, this protein was detected neither in cells nor in tumors, while changes were observed for S100A6, S100A10, and S100A11 but not for S100A8 (Appendix A). Analysis of changes in abundance between the different stages for the eight markers identified above plus S100A9 revealed a comparable evolution for S100A4, Tubulin-specific chaperone A (TBCA), and, to a lesser extent Annexin A2 (ANXA2), the former still being the most sensitive to invasion by M5-T1 tumor cells (Figure 6C).

## 3. Discussion

A characteristic feature of malignant mesothelioma is that it is a good base for studying both EMT and invasiveness, as is it characterized by the concomitant presence of epithelioid and sarcomatoid features [12]. To date, diagnosis, prognosis, and therapy are still challenging for this aggressive cancer, reliable predictive biomarkers are lacking, and a personalized therapeutic concept is urgently needed [28]. This work emphasized the value of S100A4, a potential biomarker of interest involved in all steps of experimental MM pathogenesis, which could contribute to fulfilling these aims. Moreover, the methodology and preclinical models that we used allowed us to analyze in parallel the additional value of some other proteins, whose interrelationships with S100A4 raise specific questions.

The pertinence of several MM biomarkers has been analyzed over the last decade; however, for diagnostic purposes, none are satisfactory, although many are under investigation [29,30]. Sudo et al. previously emphasized the fact that while the pathological diagnostic markers for epithelioid MM were established, no adequate marker for sarcomatoid MM was found, when they identified AHNAK as a potential marker 5 years ago [31]. Although AHNAK mesothelioma-associated gene mutation was recently found in a rare type of human MM, its role in MM pathogenesis is still under investigation [32]. Interestingly, our results obtained on experimental cell lines and tumor models of sarcomatoid MM in rats corroborated the work of Fassina et al. on a collection of 109 malignant mesothelioma specimens from patients who observed a progressive increase in S100A4 immunohistochemical staining from epithelioid to biphasic and sarcomatoid subtypes [33]. This elevation of S100A4 is also consistent with its overexpression in different types of metastatic cancer [1], and its established value as an indicator of poor prognosis and a therapeutic target for colon cancer [34].

Several studies have already been dedicated to proteomic analyses of different types of MM samples, cell lines, sera, pleural effusions, or surgically resected materials from patients. However, few have provided quantifications of biomarkers of interest identified in each situation. Among them, Ziegler et al. used cell surface capturing (CSC) technology to identify biomarkers which allowed them to differentiate between the human pleural MM cell line ZL55 and lung adenocarcinoma (ADCA) Calu-3, and they identified Thy-1/CD90 as a candidate [35]. Mundt et al. applied another technique for distinguishing pleural effusions in patients with MM, ADCA, or benign mesotheliosis, identifying galectin-1 as a diagnostic biomarker [36]. Subsequently, 142 differentially expressed proteins were identified by comparing epithelioid and sarcomatoid pleural MM samples [37]. White et al. recently extended this work using the new technology SWATH-MS, finding that protein S100A6 was upregulated 2.5-fold and 2.1-fold in MM versus ADCA and MM versus benign reactive effusions, respectively [38]. This finding is interesting as it corroborates our data on the increased abundance of S100A6 that accompanied the increased abundance of S100A4 in all three invasive MM tumors versus the non-invasive MM tumor. Finally, in the work of Creaney et al., where six human MM cell lines were compared against three primary mesothelial cell culture preparations using iTRAQ^®^ mass spectrometry (isobaric tag for relativeand absolute quantitation) [39], two upregulated proteins were also found in our data, annexin A2 and prelamin-A/C.

In our study, the changes in S100A4 within the different categories of preneoplastic rat mesothelial cell lines were consistent with our observation that *Hif1a* expression increases between PNsarc2 vs. PNsarc1. The presence of HIF-1 could also explain the parallel rise in *Vegfa* expression in these cell lines, and the highest invasiveness of M5-T1 tumor [17], in agreement with the statement that its presence in tumor microenvironment could foster the expression of VEGF, among others [5]. The relationship between HIF-1/2α and EMT is now clearly demonstrated [40], and the links between HIF-1α expression, EMT, and generation of distant metastases, for example in advanced lung cancer, are established [41]. However, we established for the first time a link between the increase in S100A4 in the context of hypoxia and EMT in the case of preneoplastic mesothelial cells. To summarize, in line with previous observations that in normal and benign lesions, S100A4 is restricted to a few stromal fibroblasts and inflammatory cells [42], while both tumor and stromal cells secrete the protein in malignant tumors [42,43], we observed a continuous rise in S100A4 from subnormal, epithelioid, sarcomatoid non-tumorigenic mesothelial cells to MM cell lines. The increased abundance of this protein in invasive versus non-invasive MM cell lines and invasive versus non-invasive MM tumors was consistent with reports of its higher expression in the peripheral leading edge of breast cancer [43], non-small-cell lung cancer (NSCLC) [44], and gain of S100A4/loss of membrane E-cadherin in cervical cancer with an unfavorable prognosis [8]. The parallel between S100A4 and fibronectin is also of interest, as the continuous increase in protein abundance between subgroups Sbnl, PNep, and PNint, and the difference between sarcomatoid and epithelioid preneoplastic rat mesothelial cell lines suggest a relationship with the Wnt signaling pathway [45]. In our study, this rise in fibronectin also observed within rat MM tumors, previously reported to be associated with increasing invasiveness [17], was comparable to that observed in the sarcomatoid vs. epithelioid human MM tumors, in good agreement with the role of Wnt signaling pathway in NSCLC [46]. Regarding the value of S100A4 compared to other known prognostic markers such as p16 and PD-L1, we previously showed a decreased relative expression of *Cdkn2a* (p16) in neoplastic relative to preneoplastic rat cell lines and in human MM cell lines relative to normal mesothelial cells [18]. Interestingly, our findings on MM were consistent with the data of Pezzuto et al., who found altered expression of p16 in patients with NSCLC, indicating that p16 could be a potential marker of lung cancer evolution and aggressiveness, unlike PD-L1 and Ki-67 which did not affect overall survival [47]. Other findings by Chapel et al. in a cohort of patients with malignant pleural mesothelioma confirmed that tumor PD-L1 expression was not significantly associated with overall survival [48]. Overall, these findings could contribute to improve the potential utility of invasiveness biomarkers, beside the role of mesothelin, osteopontin, fibulin-3, hyaluronic acid, VEGF, and MPF in the diagnosis and prognosis of MM [49].

Among the list of markers of tumorigenesis/invasiveness that were previously identified, eight proteins evolved in a similar way in the spleens of tumor-bearing rats as in control rats, of which S100A4 was the most effected. Besides the dramatic changes observed for S100A4, S100A9 was the only member of this family that was detected neither in cells nor in tumors. As this protein belongs to the list of molecular components that promote pre-metastatic niche formation [2], acting as a chemoattractant for macrophages and hematopoietic progenitor cells accumulating in pre-metastatic lungs [50,51], the changes observed could be explained by variations in cellular composition of this central lymphoid organ. With S100A8, they represent damage-associated molecular pattern molecules (DAMPs) [52], producing a complex that promotes cancer development and invasion [53], a promising target for the development of new strategies of treatment [54]. S100A4 and S100A9 show common molecular interactions [55], both being associated with inflammation and in vivo tumor progression, reducing overall survival for the patient [3]. However, the fact that S100A9 increased while S100A8 remained unchanged suggests an imbalance between the homodimer and heterodimer/heterotetramer forms mentioned in the literature [56], a finding emphasized by the lack of significant changes for S100A8 between preneoplastic and neoplastic cell lines, and between invasive and non-invasive neoplastic cell lines. The parallel evolution of TBCA and ANXA2 also raises additional questions. TBCA is a member of a family of proteins that act at different levels of a folding pathway to generate tubulin heterodimers involved in many cellular functions. As two independent studies have reported a role for these chaperones in the sensitivity/resistance to treatment of different types of cancer [57,58], the similarity of S100A4 and TBCA profiles in rat spleen in the context of MM progression is intriguing. The additional data on ANXA2 finally suggest that markers of communication between cells/intracellular transport, cell motility, and cellular architecture are closely interconnected, especially in the more advanced stages, highlighting the key role of the S100A4/annexin A2 interaction [59].

Other than S100A4, our study also led us to identify several other potential markers of interest for MM, including annexin A5 [60], as its serum level in patients with colon cancer is related to lymph node metastasis and tumor grades [61], while investigations of its role and action in hepatocarcinoma malignancy has revealed that its knockdown suppressed the expression of key molecules in the integrin and MEK-ERK pathways (mitogen-activated protein kinases-extracellular signal-regulated kinases) [62]. Regarding lamin A/C, Kim et al. demonstrated that it forms a perinuclear apical actin cap, resisting nuclear deformation in response to physiological mechanical stresses [63]. Interestingly, the continuous rise in LMNA in F4-T2, F5-T1, and M5-T1 tumors is consistent with an increased tumor invasiveness [17] and the involvement of EMT at the invasive tumor front [64]. The parallel increase in H2A.J also corroborates the findings of Taheri et al., showing that the mobility of lamin A and histone H2A are interconnected in the nucleus [65], while H2A.J plays a central role in senescent cells with persistent DNA damage and in the expression of inflammatory genes that contribute to the senescent-associated secretory phenotype (SASP) [66].

## 4. Materials and Methods

### 4.1. Biocollection of F344 Rat Mesothelial Cell Lines

The rat mesothelial biocollection of cell lines was established in 2011 from a group of 33 F344 rats, which received an intraperitoneal inoculation of 10 mg crocidolite fibers (UICC analytical sample, ref. 02704A, Neyco, Paris, France) at 8 weeks of age. The 22 preneoplastic and 4 neoplastic cell lines used in this study were obtained from individual rats from 136 to 415 days after induction, as previously described [17], and their classification is summarized in Table 1. Cell lines were grown in an RPMI 1640 medium, supplemented with 10% heat-inactivated fetal bovine serum, 2 mM L-glutamine, 100 U/mL penicillin, and 100 µg/mL streptomycin (all reagents from Gibco Life Technologies Limited, Paisley, UK) at 37 °C in a 5% CO_2_ atmosphere. Cells were collected from preconfluent 75 cm^2^ flasks of each cell line and cell pellets of 2 × 10^6^ cells were used for SWATH-MS proteomic analysis after being washed in PBS (phosphate-buffered saline) buffer.

### 4.2. Total RNA Isolation and Real-Time PCR

To complete our previous classification of the biocollection of rat mesothelial cell lines, PCR reactions were performed as previously described [18] for 15 additional genes for the nine cell lines representing the PNsarc group of preneoplastic cells (Table 1)—*Il6*, *Fgf2*, *Egf*, *Vegfa*, *Pdgfa*, *Fhit*, *Hif1a*, *Stat3*, *Erbb2*, *Nfkb1*, *Tnf*, *Il1b*, *Ccl5*, *Ccl7*, and *Ccl3*. Each transcript level was normalized relative to the acidic ribosomal phosphoprotein P0 housekeeping gene (*Rplp0*), used as an internal standard.

### 4.3. Malignant Mesothelioma Tumors

Fischer F344 rats were purchased from Charles River Laboratories (L’Arbresle, France), and maintained under SPF status and standard conditions in the UTE-IRS UN animal facility of the University of Nantes in compliance with the European Union guidelines for the care and use of laboratory animals in research protocols (Agreement #01257.03). All experiments were approved by the ethics committee for animal experiments of the Pays de la Loire Region, France (CEEA.2011.38 and CEEA.2013.7.). In order to follow the recommendations on the welfare and use of animals in cancer research, particular attention was given to incorporating the objectives of the 3Rs (replacement, reduction, and refinement) [67]. The four neoplastic cell lines (M5-T2, F4-T2, F5-T1, and M5-T1) were injected into syngeneic rats and tumors were collected and fixed as previously described [16,17]. The rats were anesthetized in an isoflurane chamber (Forene^®^, Abbott, France) and euthanized with a rate of 30% volume displacement per minute of CO_2_ into their home cage.

### 4.4. Sample Preparation for SWATH-MS Analysis

The rat spectral library, DDA experiments, and peptide identification were performed as previously described (see Materials and Methods section in Reference [17]). The staging (Ki67 index) and histology features (infiltration atypia, blood vessel density, and T cell and macrophage counts) of the four different rat MM models were described in Figure 1, Table 1 and Table 2 [17]. Paraffin-embedded sections of representative tumors, stained with hematoxylin-phloxine-saffron (HPS), were first examined to select areas of interest, and then all the corresponding areas were removed with a scalpel from five thicker (20 µm) sections of the samples and collected in a microtube. Samples were deparaffinized with xylene, subjected to a succession of incubations in 100%, 95%, 70% and 50% ethanol baths followed by centrifugations, and finally dried in a SPD121P SpeedVac Concentrator. Cell pellets and dried deparaffinized tumor samples were treated with Rapigest^TM^ SF Surfactant (Waters Corporation, Milford, MA, USA) and DTT 0.1 M at 95 °C for 90 min with continuous stirring (Thermomixer Comfort, Eppendorf), and then sonicated three times for 30 sec (ultrasonic processor, model 75185, Bioblock Scientific, Illkirch, France). MMTS 200 mM was then added for 10 min at 37 °C with continuous stirring before being treated with trypsin w/CaCl_2_ TPCK (P/N 4352157, AB Sciex Pte, Ltd., Framingham, MA, USA) overnight. After centrifugation, salts were removed using OASIS^®^ HLB extraction cartridges (Waters SAS, St Quentin-en-Yvelynes, France.) before drying under SpeedVac. Peptide concentrations of the samples were finally determined using the Micro BCA^TM^ protein assay kit (Thermofisher Scientific, St Herblain, France). For treatment with trypsin, desalting, and peptide analysis, ultrapure water (Purelab Option-Q, 18.2 MΩ.cm, VWR International S.A.S., Fontenay-sous-Bois, France) was used during all procedures. See legends to Appendix A and reference [68] for additional informations on sample preparation of human tumor samples.

### 4.5. Relative Quantification by SWATH Acquisition and Statistical Analysis

Each sample (5 µg) was analyzed with LC-MS/MS equipment and a LC gradient to build the spectral library, but using a SWATH-MS acquisition method. The method consisted of repeating the whole gradient cycle, which corresponded to acquisition of 32 TOF (time of flight) MS/MS scans of overlapping sequential precursor isolation windows (25 m/z isolation width, 1 m/z overlap, high sensitivity mode) covering the 400 to 1200 m/z mass range, with a previous MS scan for each cycle. The accumulation time was 50 ms for the MS scan (from 400 to 1200 m/z) and 100 ms for the product ion scan (230 to 1500 m/z), thus giving a total cycle time of 3.5 sec.

Peak extraction of the SWATH data was performed using either the Spectronaut software (v. 8.0, Biognosys, Switzerland) or SWATH micro App embedded in Peak View (v. 2.0, AB Sciex Pte, Ltd., Framingham, MA, USA). SWATH data were processed with default settings in Spectronaut. Reference peptides from the iRT-kit (Biognosys) spiked into each sample were used to calibrate the retention time of extracted peptide peaks using Spectronaut. Peptide identification results were filtered with a q-value of <1%, excluding shared peptides. RT calibration was also performed based on iRT peptide elution profiles in PeakView using the SWATH App module (v. 2.0). After peak extraction with either Spectronaut or PeakView, the sum of MS2 ion peak areas of SWATH quantified peptides for individual proteins were exported to calculate the protein peak areas.

For the statistical analysis of the SWATH data set, peak extraction output data matrix from PeakView was imported into MarkerView (v. 2, Sciex) for data normalization and relative protein quantification. Proteins with a fold change >1.5 and statistical *p*-value <0.05 estimated by MarkerView were considered to be differentially expressed under different conditions.

### 4.6. MRI-Based Staging of the M5-T1 MM Model

Coronal and axial MR images of M5-T1-tumor bearing rats (in comparison with normal rats) were acquired at Day 15 after tumor challenge on a Biospec Avance III 70/20 MR scanner (Bruker Wissembourg, France) working at 7.0 T, equipped with a horizontal bore magnet, 120 mm diameter gradient system (675 mT/m), and transmitter/receiver 86 mm volume coil. Data were collected using ParaVision 6.0 software (Bruker, Wissembourg, France). A fast spin echo sequence was applied with following parameters to this end: TE (Echo Time) = 30 ms, TR min = 740 ms (respiratory triggering), four averages, echo train length = 4, 60 × 60 mm field of view, 256 × 256 matrix, nine slices with a thickness of 1 mm oriented either axially or sagittally.

## 5. Conclusions

In conclusion, these results showed that S100A4 was the only protein with a common increase in abundance in all the situations studied, highlighting a crucial role for this biomarker in the multistep MM pathogenesis process, including tumorigenesis, EMT, invasion, and colonization of host organs. Its relationships with members of the annexin family, and the dynamic interplay between lamins and chromatin, represent an interesting basis for future mechanistic studies.

## Figures and Tables

**Figure 1 cancers-12-00939-f001:**
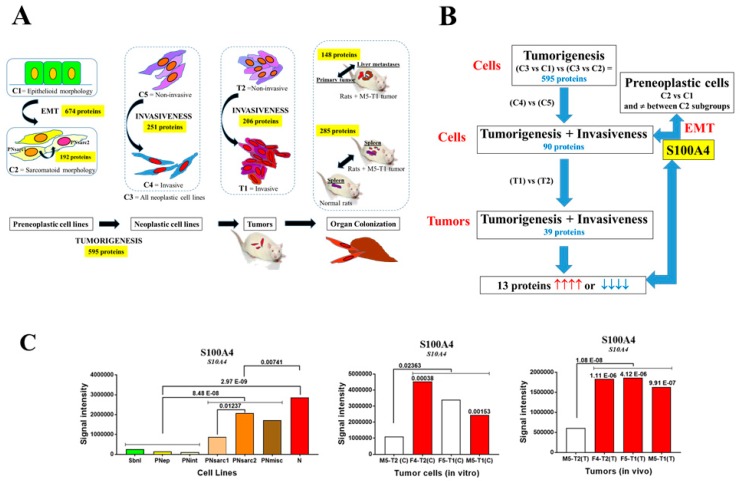
Identification of S100A4 as the main biomarker. (**A**) Scheme of the successive steps and sources of data used for the identification of candidate biomarkers. C1, preneoplastic cell lines with epithelioid morphology. C2, preneoplastic cell lines with sarcomatoid morphology. C3, neoplastic cell lines. C4, invasive neoplastic cell lines. C5, non-invasive neoplastic cell line. T1, invasive MM (malignant mesothelioma) tumors. T2, non-invasive MM tumor. (**B**) Methodology used for SWATH-MS (sequential window acquisition of all theoretical fragmentation spectra) proteomic analyses of cell lines and tumors. (**C**) Comparative proteomic abundances of S100A4 (significant increases in red and decreases in blue, with *p* values). Left: comparison between the different subgroups and groups of preneoplastic and neoplastic cell lines (in vitro); middle: comparison between invasive and non-invasive neoplastic cell lines (in vitro); right: comparison between invasive and non-invasive mesotheliomas (in vivo). Blank bars correspond to the absence of significant differences between cell lines/tumors.

**Figure 2 cancers-12-00939-f002:**
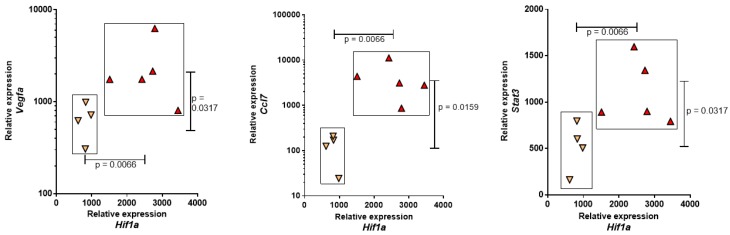
Differential expression of *Hif1a* within preneoplastic cell lines with sarcomatoid morphology (PNsarc, C2). Expression of *Hif1a* in the two subgroups (PNsarc1 in yellow, PNsarc2 in red triangles) of rat preneoplastic mesothelial cell lines with sarcomatoid morphology and their relationships with *Vegfa*, *Ccl7*, and *Stat3* expressions.

**Figure 3 cancers-12-00939-f003:**
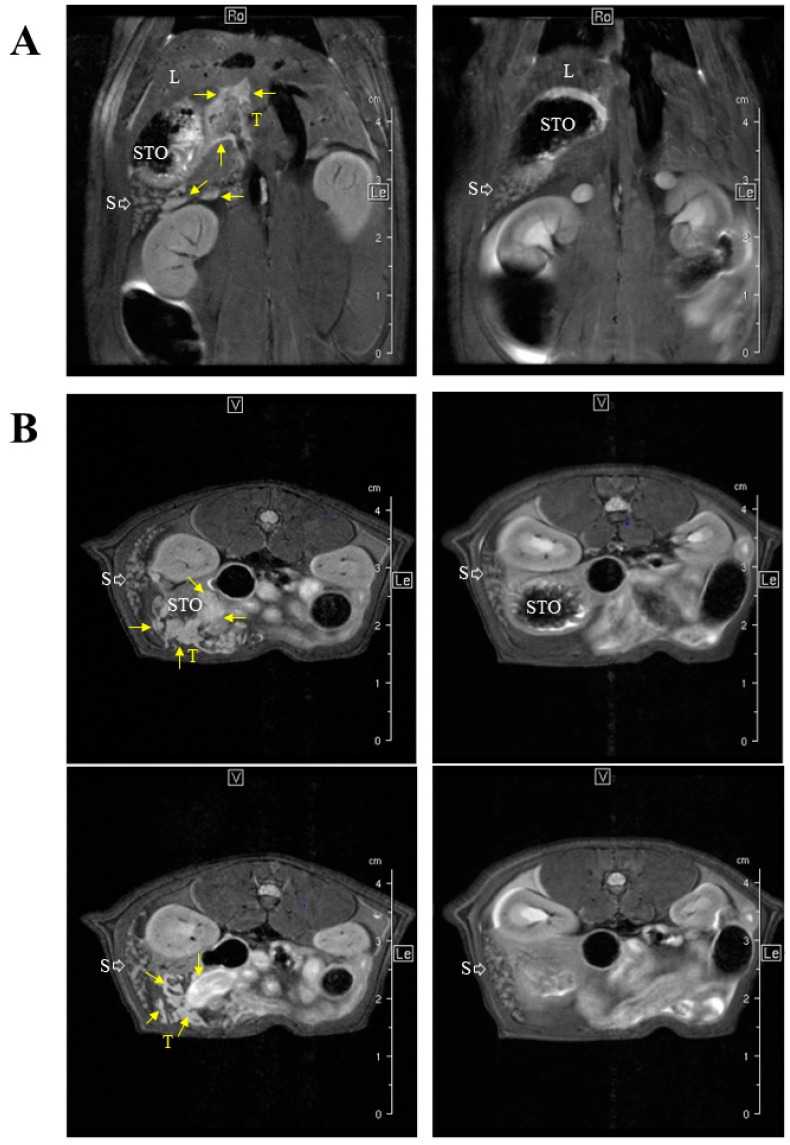
MRI-based staging of the M5-T1 MM model. M5-T1 tumor (T, and yellow arrows) was present as omental cakes and nodules attached to the stomach (STO) and spleen (S) on both coronal (**A**) and axial images (**B**). M5-T1 tumor was also characterized by the presence of metastatic tumor development into the liver parenchyma (L) and along the portal vein. Note the changes in volume and white pulp vs. red pulp signal intensity of the spleen in the tumor-bearing rat in comparison to normal rat spleen.

**Figure 4 cancers-12-00939-f004:**
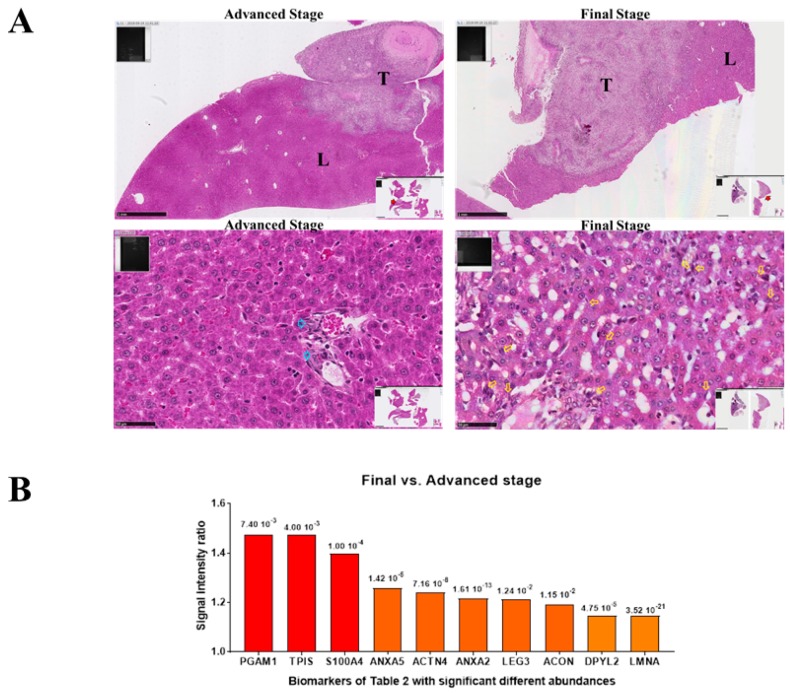
Comparison between liver metastases and primary tumor, M5-T1 tumor model. (**A**) Histological features of the two stages identified in the colonization of the liver by M5-T1 cells. Top rank, general views (×25, the scale bar represents 1 mm) showing the extent of tumor (T) development within the liver tissue (L). Bottom rank, detailed views (×400, the scale bar represents 50 µm). The final stage was characterized by the presence of numerous isolated metastatic M5-T1 cells infiltrating the liver parenchyma (yellow arrows), in contrast to the advanced stage for which the tumor cells were only localized within the portal triad (blue arrows). (**B**) Proteomic analysis: Abundance ratios (final versus advanced stage) for the 11 biomarkers exhibiting significant differences (*p* < 0.05), and also found in the list of 39 biomarkers detailed in Table 2, *p* values are indicated for each biomarker at the top of the bars.

**Figure 5 cancers-12-00939-f005:**
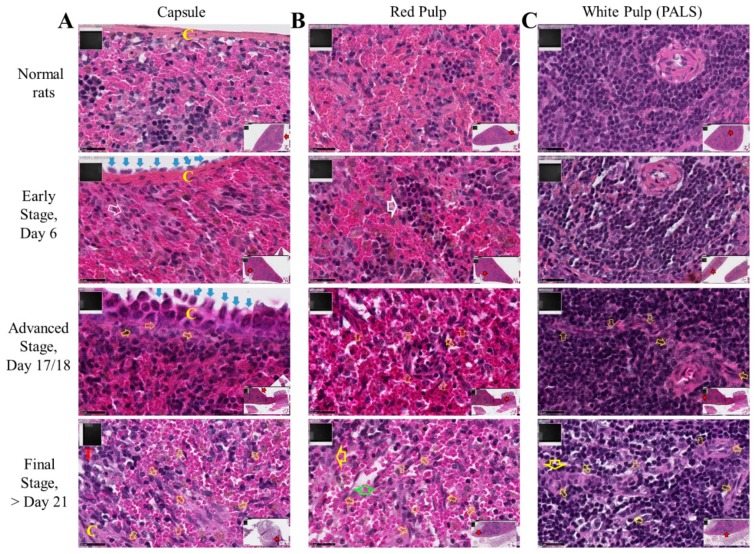
Spleen colonization by M5-T1 cells. High magnification views of the successive stages (×800). Scale bars represent 25 µm, inserts showing general views (×25). (**A**) Invasion of the capsule (C, in yellow). The open white arrow shows clustering of reticular cells. Open yellow arrows indicate M5-T1 tumor cells crossing the capsule or present in the red pulp. Vicinal tumor, T in red. (**B**) Red pulp. Early stage: the architecture of the red pulp is preserved, large open white arrow indicating clusters of lymphoid cells on the tumor side attached to the spleen surface. Advanced stage and onwards: open yellow arrows indicate numerous tumor cells actively moving inside the red pulp. Final stage: the large green arrow shows empty spaces, attesting a decrease in the density of normal cells. The large yellow arrow indicates clusters of lymphoid and tumor cells. (**C**) White pulp, periarteriolar lymphoid sheath (PALS). Early stage: the architecture of the follicle is preserved on this cross-section of the central artery. Advanced stage: open yellow arrows indicate tumor cells penetrating the PALS on this longitudinal section of the central artery. Final stage: tumor cells present everywhere following the destruction of the stromal framework of reticular fibers.

**Figure 6 cancers-12-00939-f006:**
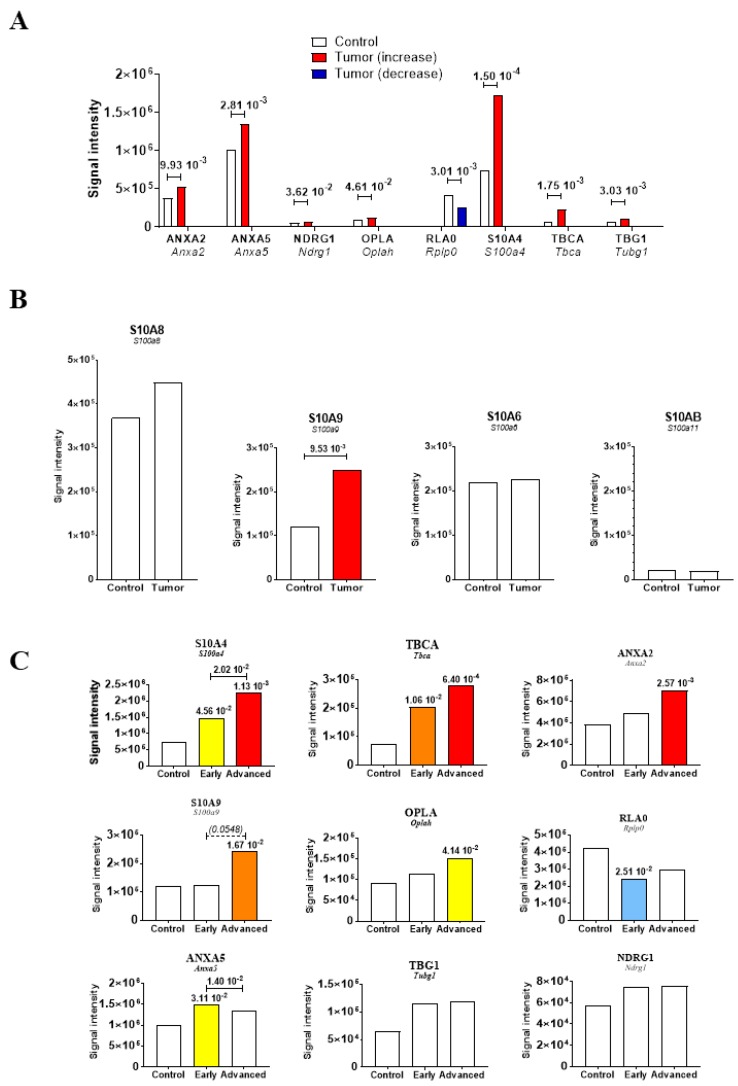
Biomarkers of spleen colonization. Changes in protein abundances in the spleen tissue of rats with M5-T1 mesothelioma (Tumor) in comparison with control rats (Control). (**A**) Eight potential biomarkers showing significant differences. Proteins detected in the spleen from the 39 proteins listed in Table 2. (**B**) Other S100 proteins detected in spleen. (**C**) Evolution of the eight biomarkers plus S100A9 according to the stage of M5-T1 mesothelioma development. “Early” and “Advanced” correspond to the spleens of tumor-bearing rats. Blank bars correspond to the absence of significant differences between groups.

**Table 1 cancers-12-00939-t001:** Classification of the biocollection of rat mesothelial cell lines. Roulois et al. [18] showed photographs of each cell line at confluence and their growth characteristics (Appendix A and Appendix A, respectively).

Category	Group	Subgroup	Name	Differential Gene Expression
**Preneoplastic** **(PN)**	**Epithelioid morphology** (PN-[Epith]) **C1**	Subnormal (Sbnl)	F1-0e	Pdpn^high^ Ezr^high^ Cdh1^high^ Acta2_low_ Myc^high^ Igf1_low_ TGFb1_low_ Il10_low_
			F1-0f	
			M2-0	
		Preneoplastic epithelioid (PNep)	F2-0	Pdpn^high^ Ezr^high^ Cdh1med Acta2_low_ Myc_low_ Igf1^high^ TGFb1_low_ Il10^high^
			F2-3	
			F2-4	
			F3-1	
			M3-1	
		Preneoplastic intermediary (PNint)	F1-2	Pdpn^high^ Ezr^high^ Cdh1med Acta2med TGFb1^high^ Il10_low_
			F5-2	
	**Sarcomatoid morphology** (PNsarc) **C2**	Sarcomatoid hypoxia low (PNsarc1)	F1-1	Cdh1_low_ Acta2^high^ Tgfb1^high^ Zeb^high^ Hif1a_low_ CCl7_low_ Vegfa_low_
			F2-2	
			M2-1	
			M3-2	
		Sarcomatoid hypoxia high (PNsarc2)	F1-3	Cdh1_low_ Acta2^high^ Tgfb1^high^ Zeb^high^ Hif1a^high^ CCl7^high^ Vegfa^high^
			F2-1	
			F5-3	
			M1-1	
			M2-3	
		Miscellaneous (PNmisc)	M2-2	Acta2med Ezrmed Tgfb1^high^
			M4-1	
			F2-5	
			F3-2	
**Neoplastic** **(N)** **C3**	Non-invasive **C5**		M5-T2	Cdkn2alow Lgals3high Vegfahigh
	Invasive **C4**		F4-T2	Cdkn2alow Lgals3high Vegfalow
			F5-T1	
			M5-T1	

**Table 2 cancers-12-00939-t002:** List of candidate markers of interest, identified via the different methodological steps described in Figure 1B. The 13 proteins for which abundances changed significantly (*p* < 0.05) in the same way in the different situations are highlighted in yellow (increase) or green (decrease).

Code	Name	C3 vs. C1	C3 vs. C2	C4 vs. C5	T1 vs. T2
ACADL	Long-chain specific acyl-CoA dehydrogenase, mitochondrial	**↓**	**↓**	**↓**	**↑**
ACON	Aconitate hydratase, mitochondrial	**↓**	**↓**	**↓**	**↓**
ACTN4	Alpha-actinin-4	**↑**	**↓**	**↓**	**↓**
ANXA2	Annexin A2	**↑**	**↑**	**↑**	**↓**
ANXA5	Annexin A5	**↑**	**↑**	**↑**	**↑**
ATPO	ATP synthase subunit O, mitochondrial	**↑**	**↑**	**↓**	**↑**
BAF	Barrier-to-autointegration factor	**↑**	**↑**	**↑**	**↑**
DPYL2	Dihydropyrimidinase-related protein 2	**↓**	**↓**	**↓**	**↓**
EDF1	Endothelial differentiation-related factor 1	**↑**	**↑**	**↓**	**↓**
EMAL2	Echinoderm microtubule-associated protein-like 2	**↓**	**↓**	**↓**	**↓**
FKB1A	Peptidyl-prolyl cis-trans isomerase FKBP1A	**↑**	**↑**	**↓**	**↓**
GSHR	Glutathione reductase	**↑**	**↑**	**↑**	**↑**
H2AJ	Histone H2A.J	**↑**	**↑**	**↑**	**↑**
H31	Histone H3.1	**↓**	**↓**	**↑**	**↑**
HYOU1	Hypoxia up-regulated protein 1	**↓**	**↑**	**↓**	**↓**
ICAL	Calpastatin	**↓**	**↓**	**↑**	**↑**
IF6	Eukaryotic translation initiation factor 6	**↓**	**↓**	**↓**	**↑**
LMNA	Prelamin-A/C	**↑**	**↑**	**↑**	**↑**
LUZP1	Leucine zipper protein 1	**↓**	**↓**	**↓**	**↓**
MOES	Moesin	**↑**	**↑**	**↓**	**↓**
MYH10	Myosin-10	**↓**	**↓**	**↓**	**↓**
NCAM1	Neural cell adhesion molecule 1	**↑**	**↑**	**↓**	**↓**
NDKB	Nucleoside diphosphate kinase B	**↑**	**↑**	**↓**	**↓**
NDRG1	Protein NDRG1	**↓**	**↓**	**↓**	**↑**
OPLA	5-oxoprolinase	**↓**	**↓**	**↓**	**↓**
PA1B2	Platelet-activating factor acetylhydrolase IB subunit beta	**↑**	**↑**	**↓**	**↓**
PHB	Prohibitin	**↓**	**↓**	**↑**	**↑**
PRDX3	Thioredoxin-dependent peroxide reductase, mitochondrial	**↑**	**↑**	**↓**	**↓**
RLA0	60S acidic ribosomal protein P0	**↑**	**↑**	**↑**	**↓**
RTCB	tRNA-splicing ligase RtcB homolog	**↓**	**↓**	**↓**	**↑**
S10A4	Protein S100-A4	**↑**	**↑**	**↑**	**↑**
SEPT2	Septin-2	**↓**	**↓**	**↓**	**↓**
STABP	STAM-binding protein OS = Rattus norvegicus	**↓**	**↓**	**↓**	**↑**
TBCA	Tubulin-specific chaperone A	**↓**	**↓**	**↑**	**↑**
TBG1	Tubulin gamma-1 chain	**↓**	**↓**	**↑**	**↑**
TCTP	Translationally-controlled tumor protein	**↑**	**↑**	**↓**	**↓**
TYB10	Thymosin beta-10	**↑**	**↑**	**↓**	**↓**
UB2V2	Ubiquitin-conjugating enzyme E2 variant 2	**↑**	**↑**	**↓**	**↓**
UGGG1	UDP-glucose:glycoprotein glucosyltransferase 1	**↓**	**↓**	**↑**	**↑**

**Table 3 cancers-12-00939-t003:** Abundances of candidate biomarkers of tumor cell invasiveness in the spleens of rats with M5-T1 tumors relative to the spleens of control rats.

Code	Name	Rats with Tumors vs. Control Rats
ACADL	Long-chain specific acyl-CoA dehydrogenase, mitochondrial	ns
ACON	Aconitate hydratase, mitochondrial	ns
ACTN4	Alpha-actinin-4	ns
ANXA2	Annexin A2	**↑**
ANXA5	Annexin A5	**↑**
ATPO	ATP synthase subunit O, mitochondrial	ns
BAF	Barrier-to-autointegration factor	ns
DPYL2	Dihydropyrimidinase-related protein 2	ns
EDF1	Endothelial differentiation-related factor 1	nd
EMAL2	Echinoderm microtubule-associated protein-like 2	ns
FKB1A	Peptidyl-prolyl cis-trans isomerase FKBP1A	ns
GSHR	Glutathione reductase	nd
H2AJ	Histone H2A.J	ns
H31	Histone H3.1	ns
HYOU1	Hypoxia up-regulated protein 1	ns
ICAL	Calpastatin	ns
IF6	Eukaryotic translation initiation factor 6	ns
LMNA	Prelamin-A/C	(↑) *p =* 0.05103
LUZP1	Leucine zipper protein 1	ns
MOES	Moesin	ns
MYH10	Myosin-10	ns
NCAM1	Neural cell adhesion molecule 1	nd
NDKB	Nucleoside diphosphate kinase B	(↑) *p =* 0.06241
NDRG1	Protein NDRG1	**↑**
OPLA	5-oxoprolinase	**↑**
PA1B2	Platelet-activating factor acetylhydrolase IB subunit beta	ns
PHB	Prohibitin	ns
PRDX3	Thioredoxin-dependent peroxide reductase, mitochondrial	ns
RLA0	60S acidic ribosomal protein P0	**↓**
RTCB	tRNA-splicing ligase RtcB homolog	ns
S10A4	Protein S100-A4	**↑**
SEPT2	Septin-2	ns
STABP	STAM-binding protein OS=Rattus norvegicus	ns
TBCA	Tubulin-specific chaperone A	**↑**
TBG1	Tubulin gamma-1 chain	**↑**
TCTP	Translationally-controlled tumor protein	ns
TYB10	Thymosin beta-10	(↓) *p =* 0.05936
UB2V2	Ubiquitin-conjugating enzyme E2 variant 2	ns
UGGG1	UDP-glucose:glycoprotein glucosyltransferase 1	ns

The eight main proteins of interest are highlighted with arrows indicating the direction of change. Three additional proteins of interest showed strong tendencies (0.050 < *p* <0.063). ns = differences not significant (*p* > 0.07), nd = proteins not detected in spleen samples, yellow = increase, green = decrease.

## Data Availability

All informations about the data and materials used are included in the article or the accompanying additional files.

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
