# Peer review of "S100A4 Is a Biomarker of Tumorigenesis, EMT, Invasion, and Colonization of Host Organs in Experimental Malignant Mesothelioma"

_cancers, 2020, doi:10.3390/cancers12040939_

Round 1

Reviewer 1 Report

I suggest some changes to improve the paper that is interesting.

Cancers
Title of Manuscript: S100A4 is a biomarker of tumorigenesis, EMT, 2 invasion and colonization of host organs in 3 experimental malignant mesothelioma.
I recommend minor revision

COMMENTS
It is a valuable and interesting study that focused on the role of S100A4 protein in in the cross-talking between microenvironment and cancer cells in metastatic mesothelioma.
Some missing data should be added.
I suggest to add some data that could improve the discussion
Moreover further specifications chould be done to clarify the topic.
About the mass spectrometry analysis: how the protein panel was chosen? Why have no other proteins potentially oncogenic and anti-oncogenic been selected?
In section 4.4 please clarify the staging and histology features of considered paraffin-embedded sections.
I suggest to include section on possible perspective derivinf from these findings.
Some references should be added and discussed for comparison:
- Anticancer Res 2020;4(2):983-90 for comparison with p16 and its prognostic and anti-oncogene action.
- Oncotarget 2019;10(66):7071-79 for comparison with HIF that has proven to be a prognostic factor in resected lung cancer patients.
- J Natl Cancer Inst. 2014 Jan;106(1):djt356 about WNT signaling and lung cancer, the link with cell proliferation and apoptosis
- Ann Clin Biochem. 2018 Jan;55(1):49-58 osteopontin, fibulin-3 and vascular endothelial growth factor are independently associated with poor prognosis
- Hum Pathol. 2019 May;87:11-17 about the prognostic role of PD-L1 in mesothelioma

Reviewer 2 Report

Authors of this article investigated differential expression of proteins in the collection of experimental preneoplastic and neoplastic rat mesothelioma cell lines by using proteomic approach. Their analysis found the S100A4 to be upregulated in neoplastic tissue compared preneoplastic and in the invasive vs. non-invasive suggesting this protein to be biomarker for invasive and sarcomatoid type of mesothelioma. This study is relevant due to poorly understood biology of mesothelioma.

Minor revision:

1) In Table 1, additional label of C1/C2/C3/C4/C5 groups (which connect to Figure 1) would help to understand easier the analysis.

2) Figure 2, is poorly explained within the scope of the whole study. Could authors provide more explanation of the figure 2 the study? How expression of HIF1 and Vegfa, Ccl7 and Stat3 contributes with the finding of S100A4?

3) Figure 4 line 167, assumed typo mistake “5.T1”

Major revision:

Major remark to this study would be that all analysis were only performed by using proteomics. The findings could be supported via additional experiments.

This is important particularly important within clinical context, where a potential S100A biomarker will be accessed rather by any immunoassays. Thus, additional experiments using western blot or immunofluroscence detecting S100A4 would be necessary.

1) In Figure 1, a sample representing each group could be analysed for the presence of S100A4 by western blot.

2) What is the expression of S1004A in human mesothelioma cell lines, e.g. epitheliod vs sarcomatoid?

3 It is still debatable if the epithelioid type undergoes EMT and gives rise to sarcomatoid type of mesothelioma. To support increase in S1004A between preneoplastic epithelioid vs sarcomatoid,  is there also concominatly upregulation of any EMT markers?  

4In figure 4a, can additional immunostaining of S100A4 be performed to support findings from the proteomic screen? Since the expression of S100A is associated with the EMT, additional staining of any EMT markers would support the finding.

Reviewer 3 Report

The present manuscript (#754731) is about the S100 protein S100A4, suggesting that “S100A4 is a biomarker of tumorigenesis, EMT, invasion and colonization of host organs in experimental malignant mesothelioma”. The authors analysed a collection of 26 preneoplastic and neoplastic rat mesothelial cell lines and malignant mesothelioma models with increased invasiveness by using the SWATH-MS thechnique. Furthermore, they performed proteomic and histological analysis of sections from liver metastasis embedded in paraffin and the spleen of tumor-bearing rats, transplanted with the most invasive tumor cell model (M5-T1). The authors found that S100A4 serves as a biomarker for neoplastic transformation, EMT and invasiveness in vitro and in vivo. Additionally, according to their development stages levels of S100A4 were found to be increased in liver metastasis of malignant mesothelioma. The authors also provide evidences that S100A4 shows significantly different abundances in the spleen of malignant mesothelioma.

Overall, this study is very well performed and needs only a few corrections.

Minor concerns:

  1. Line 199, in context to S100A9, the authors state “In contrast, this protein was detected neither in cells nor in tumors”, but in the sentence before, the authors indicate that S100A9 is the only protein which was found to be increased in the spleen of tumor-bearing rats.

The statement of the authors sounds mistakable and should be executed more precisely. The authors repeat this equivocality in the discussion (line 286, 287);

  1. The authors identified to S100A4 several other S100 proteins, as S100A6, S100A8, S100A9, S100A10 and S100A11. My question is, if the authors found also further S100 proteins in their analysis or do they only identified those S100 proteins which they provide with their manuscript?

Round 2

Reviewer 2 Report

thanks for addressing the given remarks